# Variations in the Major Nutrient Composition of Dominant High-Yield Varieties (HYVs) in Parboiled and Polished Rice of Bangladesh

**DOI:** 10.3390/foods12213997

**Published:** 2023-11-01

**Authors:** Kazi Turjaun Akhter, Habibul Bari Shozib, Md. Hafizul Islam, Sneha Sarwar, Md. Mariful Islam, Md. Rubel Akanda, Muhammad Ali Siddiquee, Md. Mohiduzzaman, Abu Torab M. A. Rahim, Nazma Shaheen

**Affiliations:** 1Institute of Nutrition and Food Science, University of Dhaka, Dhaka 1000, Bangladesh; kazi.turjaun@du.ac.bd (K.T.A.); hafizinfs19@gmail.com (M.H.I.); snehasarwar@du.ac.bd (S.S.); mohiduzzaman.md@gmail.com (M.M.); torabrahim@du.ac.bd (A.T.M.A.R.); 2Bangladesh Rice Research Institute (BRRI), Gazipur 1701, Bangladesh; shozib11@gmail.com (H.B.S.); rupak115@gmail.com (M.M.I.); rubelakanda494@gmail.com (M.R.A.); mastpgpbd@yahoo.com (M.A.S.)

**Keywords:** rice varieties, parboiling, polishing, nutrient composition, Bangladesh

## Abstract

Rice (*Oryza sativa* L.) is the principal staple food, a fundamental component of food security, a significant source of energy and major nutrients, and a key player in the overall nutritional status in Bangladesh. Parboiling is a common rice-processing treatment in Bangladesh. Recently, polishing has also become a common practice among millers seeking to attract consumers. Polishing may influence the nutrient composition of rice. The present study aimed to investigate the impact of parboiling and polishing on the nutritional content of the five High Yield Varieties (HYVs) of rice (BR11, BRRI dhan28, BRRI dhan29, BRRI dhan49, and BRRI dhan84) and their percent contributions to the Recommended Dietary Allowances (RDA) of vitamins and minerals. All of the rice samples were analyzed for proximate parameters, vitamins (B1, B2, B3, B6, and folate), and minerals (Ca, Mg, Fe, Zn, Na, K, P). Moisture, ash, fat, and total dietary fiber (TDF) were determined gravimetrically, according to the AOAC Official Methods; protein was measured by the Kjeldahl method; B-group vitamins were measured using Ultra Pressure Liquid Chromatography; and mineral content was determined by ICP-OES. The energy, protein, fat, and total dietary fiber (TDF) content of the samples ranged between 342–357 kcal/100 g, 6.79–10.74 g/100 g, 0.31–1.69 g/100 g, and 2.59–3.92 g/100 g respectively. Thiamin, riboflavin, niacin, pyridoxin, and folate content ranged from 0.11–0.25 mg/100 g, 0.01–0.05 mg/100 g, 2.82–6.42 mg/100 g, 0.12–0.30 g/100 g, and 5.40–23.95 g/100 g respectively. In a comparison of parboiling and polishing, macronutrients and vitamin retention were higher in parboiled unpolished rice than in polished unparboiled rice. The minerals (mg/100 g) Ca, Mg, Fe, Zn, Na, K, and P were in the ranges 32.82–44.72, 30.69–58.34, 0.51–0.70,1.83–2.79, 5.00–5.36, 106.49–112.73, and 162.23–298.03. Minerals of BRRI dhan84 were unaffected by polishing and parboiling. BRRI dhan84 contributed a higher percentage of RDA of all B vitamins and minerals. Therefore, to reduce nutrient loss in rice, industries and consumers should be encouraged to avoid polishing or limit polishing to 10% DOM and to consume unpolished rice, either parboiled or unparboiled.

## 1. Introduction

Rice (*Oryza sativa* L.) is the principal staple food and a fundamental component of food security in Bangladesh. According to the recent Household Income and Expenditure Survey (HIES-2022), the per capita intake of rice is around 329 g daily in Bangladesh [1]. Thus, rice is a significant source of energy and vital nutrients and a key player in the nation’s overall nutritional status. As a result of favorable geographic features, environmental adaptability, climate conditions, and lower risk of regional variation, farmers favor the production of rice over other crops [2,3]. Bangladesh, with its small land area, is placed third worldwide in rice production and consumption, behind China and India [2]. However, the indigenous rice varieties have lower yields and are less resilient to climate change compared to the high-yielding varieties (HYVs) developed by the Bangladesh Rice Research Institute (BRRI) [2,4,5]. The HYVs offer shorter growth periods, have improved heat tolerance and salt tolerance, and possess multiple pest-resistance capacities [5]. In response to the growing consumption needs of the population, the government is actively promoting the adoption of HYVs. Therefore, the majority of farmers are now replacing their traditional rice varieties with HYVs. Each rice variety has its own unique characteristics and nutritional composition [6]. Additionally, value addition to the rice crop by processing methods such as parboiling and polishing, which are frequently used in rice production and preparation, might affect the nutritional value of rice [7,8].

Polished rice is mostly preferred over unpolished rice in Bangladesh. Milling is a crucial post-production step for obtaining polished rice. When rice grains are polished, the outer layer, germ, or bran is removed from the underlying starchy endosperm, resulting in a white appearance [9,10]. By lowering the likelihood of rancidity, this technique not only improves the aesthetic appeal of rice but also increases its shelf life [10]. However, as nutrients are concentrated in the bran layers of the grain, removal of the bran results in nutrient losses from the grain [3]. Therefore, although polishing improves the physical, sensory, and storage properties of rice, it also diminishes nutritional value [11].

Parboiling, another important industrial processing technique, includes three additional steps: soaking, pressure steaming, and drying before dehusking [9,12]. The transfer of nutrients from the bran to the inner layers of the rice during the soaking process is one of the main effects of parboiling, along with the inactivation of lipase enzyme from heat exposure and starch gelatinization [9,12]. Parboiling causes high nutrient retention and improves the bioavailability of vitamins and minerals; therefore, parboiled rice is considered to have superior nutritional value [11,13]. Moreover, due to enzyme inactivation, parboiled rice has an extended shelf life. The physical structure of the grain is changed during this process, which makes it resistant to breakage during milling and increases its desirability [9].

Taken together, parboiling and polishing are believed to have a considerable impact on the nutritional composition of the rice. To ascertain the nutritional composition of food, commonly referenced databases such as the USDA National Nutrient Database and national equivalents like the Food Composition Table for Bangladesh (FCTB) are frequently employed [13]. Nevertheless, these databases often lack consideration of varietal disparities. It is anticipated that different HYVs may exhibit distinct responses to analogous processing treatments, such as parboiling and polishing. Consequently, for a more comprehensive understanding, it is imperative to account for varietal differences in order to achieve precision in nutritional assessments. 

The present study aimed to investigate the impact of parboiling and polishing on the nutrient content of the five HYVs of rice in Bangladesh. This study provides extensive nutrient profiling, encompassing the proximate composition, vitamins, and mineral content of the selected dominant HYVs. It also examined the percent contribution to daily Recommended Dietary Allowances (RDA) of vitamins and minerals, especially those nutrients that have public-health importance in Bangladesh. It is anticipated that the study’s findings, which highlight potential variations in important macronutrients and micronutrients among rice varieties processed in various ways, will help with the establishment of appropriate policies to reduce the loss of nutrients due to excessive polishing of rice.

## 2. Materials and Methods

### 2.1. Sample Collection and Processing

Five HYVs, namely BR11, BRRI dhan28, BRRI dhan29, BRRI dhan49, and BRRI dhan84, were collected from the Genetic Resources and Seed (GRS) Division Bank of Bangladesh Rice Research Institute (BRRI). Among these varieties, three (BRRI dhan28, BRRI dhan29, BRRI dhan84) were Boro rice (planted in November–December and harvested in March–April) and two (BR11, BRRI dhan49) were Transplanted Aman rice (planted in March–April and harvested in November–December). Four types of sample (unparboiled unpolished, unparboiled polished, parboiled unpolished, and parboiled polished) were prepared for each of the five rice varieties. Unpolished rice is also referred to as brown rice. To obtain parboiled grains, 300 g of the rough rice (with the hulls) was placed into a beaker and one liter of distilled water was added to the beaker. The second step of the parboiling procedure involved autoclaving the hydrated rough rice for 10 min at 108 °C. Rough rice that had been autoclaved and watered was left out overnight at room temperature. Finally, a 38 °C oven dryer was used to dry the rough rice until it had a 13% moisture content. All the rice samples were dehusked using a Satake Rice Mill (THU35B, Hiroshima, Japan). To obtain polished rice, the dehusked rice grains were placed in a Grainman rice polisher (Miami, FL, USA) and polished to a 10% degree of milling (DOM) in a procedure based on a previous study [6,9] at GQN Division, BRRI. The DOM was determined using the following equation:Degree of milling (%) = (Weight of total milled rice/Weight of brown rice) × 100

The same procedure was employed to dehusk and polish the dried unparboiled and parboiled rough rice grains. The samples were then crushed with liquid nitrogen to convert them into powder form. Powdered samples were kept in a −80 °C freezer at Rice Analytical Laboratory (RAL), BRRI for further analysis, including analysis of water-soluble vitamins and minerals. 

### 2.2. Proximate Analysis

According to the ‘Official Methods of Analysis of AOAC International’ (AOAC, 2012), the content of moisture (AOAC 934.01), ash (AOAC 942.05), and fat (AOAC 963.15) were determined gravimetrically [14]. The Kjeldahl method was used to calculate total nitrogen, and the result was multiplied by 5.95 (a conversion factor specifically used for rice) for protein estimation [14]. The total dietary fiber (TDF), which is the sum of the soluble (SDF) and insoluble (IDF) fiber, was calculated using the enzymatic gravimetric method (AOAC 991.43). The content of available carbohydrates in the rice samples was calculated by subtracting the summed percentage of moisture, protein, fat, ash, and total dietary fiber. To calculate total energy content, the energy-yielding components such as crude protein, fat, fiber, and available carbohydrates were multiplied by the appropriate Atwater conversion factors [15].

### 2.3. Water-Soluble B-Vitamin Analysis

The analysis of water-soluble B vitamin (thiamine, riboflavin, niacin, pyridoxine, and folate) levels in the rice samples was carried out using Ultra-Pressure Liquid Chromatography (UPLC). A Shimadzu Prominence-i and LC-2030C 3D Plus Liquid Chromatograph were used for vitamin analysis. These water-soluble vitamins were extracted from the rice powder by acid hydrolysis followed by enzymatic hydrolysis. Briefly, 2 g of the powdered rice samples were weighed into a 250 mL Erlenmeyer flask (in triplicate). Acid hydrolysis was carried out by adding 40 mL of 0.1 M HCl and heating the mixture at 90 °C for 30 min. Thereafter, enzymatic hydrolysis was conducted by adding taka-diastase and heating at 50 °C for 2 h. After cooling, the samples were placed in tubes and centrifuged at 4000 rpm for 5 min. The supernatant was filtered through a 0.45-µm PVDF syringe filter and collected into HPLC auto-sampler vials. The aqueous extract was injected into a reverse-phase C_18_ HPLC column. The fluorescence of riboflavin was measured, and thiamin content was determined after post-column derivatization with alkaline potassium ferricyanide, which converts the thiamin to thiochrome. A modified method was used to measure thiamin and riboflavin in the rice by HPLC, as described in the ASEAN Manual of Food Analysis [16].

All extracts were analyzed using the same chromatographic technique for the B3 vitamin (as nicotinamide) and the B6 vitamin [17,18], and the triplicate extraction of each sample was prepared for analysis. All vitamin standards were purchased from Sigma Aldrich (thiamine hydrochloride, vit B1; riboflavin, vit B2; niacin, vit B3; pyridoxine, vit B6; folic acid, vit B9) (Figure 1).

### 2.4. Estimation of Minerals 

Calcium (Ca), magnesium (Mg), iron (Fe), zinc (Zn), sodium (Na), potassium (K), and phosphorus (P) were measured in mg/Kg (ppm) using a Shimadzu ICP-OES 9820 and later were converted to mg/100 g of edible portion. Rice samples were digested using a Milestone microwave digestor at RAL, BRRI. The dried rice-powder sample (1.0 gm) was taken from a plastic zip-lock package and weighed using an analytical balance. The rice sample was placed into a microwave vessel with a mixture of nitric acid and hydrogen peroxide in a 6:1 ratio (6 mL of HNO_3_ and 1 mL of H_2_O_2_). The microwave-digestion program was carried out with parameters shown in Appendix A. Next, the digested contents were transferred into polypropylene vials and topped up so total digestion could be achieved. The analysis was carried out using a Shimadzu ICP-OES 9820 coupled with a mini plasma torch, concentric nebulizer, and cyclonic chamber. The detailed instrument configuration and operation parameters for analysis are summarized in Appendix A.

To create calibration curves, concentrated nitric acid (65%), and hydrogen peroxide (30%) were used in trace-metal analysis. Ultrapure water was obtained from the Milli-Q water purification system. External calibration standards were prepared by mixing single-element stocks from Merck (Rahway, NJ, USA) and Sigma-Aldrich (St. Louis, MO, USA). Calibration standards were prepared in 1% nitric acid (Table 1). Periodic table mix 1 for ICP (TraceCERT) CRM was used as a standard solution.

### 2.5. Statistical Analysis

Analytical values of proximate composition, vitamins, and mineral composition for the samples were derived in triplicate and presented as mean, and standard deviation (SD). One-way analysis of variance (ANOVA) was used to identify differences across treatments, and Tukey’s means comparison test was applied with a significance threshold of 5%. Statistical Package for the Social Sciences (SPSS) version 25 was used for data analysis.

## 3. Results and Discussion

### 3.1. Rice Varieties and Their Production Yield

In this study, BRRI dhan28 and BRRI dhan29 are reported to have the highest yields and therefore to be consumed by the majority of the population (Appendix A). 

### 3.2. Effects of Parboiling and Polishing on the Proximate Composition of Different Rice Varieties Analyzed

Variations in the proximate composition of different dominant rice varieties due to parboiling and polishing (10% DOM) are presented in Figure 2. The energy, protein, fat, and TDF content of the analyzed rice samples ranged between 342–357 kcal/100 g, 6.79–10.74 g/100 g, 0.31–1.69 g/100 g, and 2.59–3.92 g/100 g of edible portion, respectively.

Regardless of parboiling and polishing status, all of the rice varieties had a moisture content of 10–12% (Appendix A). According to the literature, the moisture content of rice should be below 14% in order to prevent insect infestation and microbiological decomposition, particularly fungal decomposition, and to permit safe storage [13]. The moisture content of the rice varieties that were analyzed was found to be within the recommended range.

According to the findings, compared to brown rice (unpolished), polished rice had low energy, protein, and available carbohydrate content. For example, the polishing process significantly reduced the percent protein content in BR11 (9–15% reduction), BRRI dhan29 (3–10% reduction), BRRI dhan49 (10–15% reduction), and BRRI dhan84 (10–14% reduction) (Figure 2 and Appendix A). Previous research yielded comparable results regarding the effects of milling on Brazilian rice samples. Specifically, brown rice had a protein content of 6.85%, which decreased to 6.66% after milling [13]. Another study showed that the protein content of brown rice (9.2%) decreased with increasing DOM, indicating a decline in protein content from the rice surface to the endosperm; furthermore, the continued decrease after bran removal implies non-uniform distribution, with lower concentrations in the core endosperm compared to the outer layers [10]. In terms of fat content after milling, the fat content decreased drastically; the greatest decrease was seen in BRRI dhan29, in which fat content decreased up to 69–80%, followed by BR11 (72–78%), BRRI dhan49 (72–75%), BRRI dhan84 (40–72%), and BRRI dhan28 (59%). This finding is in line with earlier findings, in which the fat content decreased from 2.65% in brown rice to 0.50% in milled rice [13]. A reduction of about 62% in the fat content of rice (post-milling) was observed in another study [11]. Rice milling aims to eliminate the oil-rich aleurone layer, preventing rancidity during storage and increasing consumer appeal [11]. Most of the fat is distributed in the bran and outer endosperm, and these parts are removed during polishing to decrease the roughness of rice [13,19]. Therefore, a decrease in fat content can be observed in polished rice compared to unpolished rice. As a consequence of this fat reduction, milling reduces the fatty-acid content of the rice [11].

Based on this study, parboiling (without polishing) leads to high nutrient (energy, protein, fat, ash, available carbohydrate, and total dietary fiber) retention (Figure 2 and Appendix A). Parboiling causes protein to mobilize from the bran to the inner layer by disrupting the fibers of the grain, resulting in a greater percentage of protein in the final processed grain [9]. This result indicates that parboiled rice varieties contain more protein compared to unparboiled rice. In terms of fat, during parboiling, detached oil bodies leach out due to increased temperature and pressure, increasing the fat content of the rice [7,9].

However, parboiling caused high nutrient retention compared to polishing of unparboiled rice (Figure 2). The several factors that affect the nutrient content of rice varieties are the soaking time, the stage of germination, and the synthesis of enzymes, which is related to the production of amino acids during protein synthesis [7,8]. Therefore, any difference in any of these factors among the samples could account for the increased nutrient content of parboiled rice. However, when polishing and parboiling were undertaken together, overall nutrient retention was somewhat similar to that of non-parboiled polished rice.

The ash content of the rice varieties ranged between 0.4–1.48 g/100 g. It seems that polishing severely reduced the ash content in all rice varieties, viz., BR11 (61–70%), BRRI dhan49 (52–66%), BRRI dhan29 (47–53%), BRRI dhan84 (48–51%), and BRRI dhan28 (42–52%). An approximately 42% decrease in ash content upon milling has been reported [11]. As the ash is predominantly concentrated in the germ portion of rice, milling leads to a reduction in total ash content [11]. In contrast, parboiling increased the ash content in rice (Appendix A). Parboiling causes the migration of nutrients from the bran to the inner layer of the grain. Therefore, parboiling causes high nutrient retention. The findings of the present study strongly substantiate the findings of earlier studies [13]. When polishing and parboiling are applied concurrently, the collective mineral retention surpasses that of exclusively polished rice but falls short of the levels observed in solely parboiled rice. Thus, polishing, a practice undertaken to enhance consumer appeal, augments nutrient availability when it is combined with parboiling.

### 3.3. Parboiling and Polishing Effect on the Mineral Contents of Different Rice Varieties

Figure 3 presents the effects of different processing conditions on the mineral content of different rice varieties in Bangladesh. The levels of minerals, viz. Ca, Mg, Fe, Zn, Na, K, and P ranged within 32.82–44.72 mg/100 g, 30.69–58.34 mg/100 g, 0.51–0.70 mg/100 g, 1.83–2.79 mg/100 g, 5.00–5.36 mg/100 g, 106.49–112.73 mg/100 g, and 162.23–298.033 mg/100 g, respectively. A recent study conducted in Bangladesh highlighted varietal distinctions in the overall mineral content across 35 HYVs of rice [6]. The present study showed lower levels of thiamin, riboflavin, and iron content but a higher level of zinc content compared to earlier studies [6].

The present study examined the effect of parboiling and polishing. Brown rice, in comparison to unparboiled polished rice varieties, had higher mineral content. The levels of Ca, Mg, K, and Fe sharply decreased in polished compared to unpolished unparboiled rice (Appendix A). Overall zinc content was not drastically affected by polishing. The decrease was not uniform across varieties. For instance, polishing reduced the Mg content of BR11 (reduction by 21–22.5%), BRRI dhan29 (reduction by 4.8–5.7%), and BRRI dhan84 (reduction by 14–15%). Polishing significantly reduced Fe content in unparboiled BR11 (by 7.3%), unparboiled BRRI dhan28 (by 7.2%), and parboiled BRRI dhan29 (by 3.4%). Moreover, polishing significantly reduced P content in BR11 (by 62% in unparboiled and 48% in parboiled), and BRRI dhan84 (by about 4% in unparboiled and parboiled) varieties. Polishing increased the Na content, but not at a significant level (Appendix A).

Between the rice bran and endosperm, the majority of the minerals are found in the bran (61.0%), followed by the outer endosperm (23.7%), core endosperm (11.6%), and middle endosperm (3.7%), respectively [10]. As these elements were found to be present at high levels in the external layers of the rice samples, milling, which leads to partial removal of the external layers, causes a drop in their levels [13,20]. Earlier reports revealed that major macro elements, for instance, Ca, Mg, K, and P, undergo significant loss upon milling [13]. Other evidence showed, interestingly, that mineral loss due to milling follows an order, i.e., iron > manganese > potassium > calcium > magnesium > zinc [3]. However, none of these patterns was seen exactly in any of the rice varieties examined in the present study. These differences might be due to non-uniform mineral distribution within the grain. The mineral content of rice varieties is significantly influenced by factors such as the availability and uptake of nutrients from the soil, agricultural practices, varietal distinctions, and processing conditions. These factors collectively contribute to the observed variations in mineral content between native and non-native rice varieties [13]. However, in line with the earlier studies, overall low mineral content in polished rice was observed. A low degree of milling is generally associated with elevated mineral content. Therefore, it is advisable to prioritize the consumption of rice with a lower degree of polishing (<10% Degree of Milling or DOM) over excessively polished varieties (≥10% DOM) [6].

### 3.4. Parboiling and Polishing Effect on the Vitamin Content of Different Rice Varieties of Bangladesh

Figure 4 presents the effect of parboiling and polishing on the vitamin content of five different dominant rice varieties in Bangladesh. Thiamin, riboflavin, niacin, pyridoxine, and folate content in the analyzed samples ranged from 0.11–0.25 mg/100 g, 0.01–0.05 mg/100 g, 2.82–6.42 mg/100 g, 0.12–0.30 g/100 g, and 5.40–23.95 g/100 g respectively. Thiamin content was notably high in all samples of BRRI dhan49, ranging from 0.21–0.26 g/100 g; riboflavin (0.042–0.046 mg/100 g) and niacin (5.573–6.42 mg/100 g) levels were also high in the same variety. Pyridoxine (0.141–0.181 mg/100 g) and folate (19.277–23.953 mcg/100 g) levels were high in BRRI 84. Most of the varieties examined in the present study showed vitamin content comparable to earlier findings [21] (Appendix A).

This study also examined the effect of parboiling and polishing on vitamin content. When brown rice (unparboiled unpolished rice) was compared with unparboiled polished rice, it was observed that thiamin, riboflavin, niacin, pyridoxine, and folate content were decreased significantly upon polishing, compared to unparboiled unpolished rice varieties (Figure 4 and Appendix A). For instance, polishing reduced the content of pyridoxine in all rice varieties, notably in BRRI dhan29 (47–53% reduction) and BRRI dhan28 (21–38% reduction) (Appendix A). Earlier literature suggests that milling causes high losses of B vitamins [3]. Folate in rice was found to decrease by 72% upon milling [11]. When rice was only parboiled without polishing, vitamin retention was generally high compared to polished unparboiled rice. However, when the same rice underwent both polishing and parboiling, the vitamin content showed a pattern similar to that observed in unparboiled polished rice. Thus, the vitamin content, in descending order, is as follows: unparboiled unpolished > parboiled unpolished > unparboiled polished ≥ parboiled polished.

In this study, parboiling (without polishing) resulted in substantially higher nutrient retention than polishing. However, when parboiling and polishing were undertaken in combination, the effect was somewhat similar to that of polishing only. The findings differed across varieties (Figure 2). In parboiling, the amount of nutrient transfer from the surface layers to the center of the grain determines how effective parboiling is at reducing milling losses [22]. Even in the case of parboiling, the effects of the process were found to be sporadic in the analyzed samples. The extent to which a nutrient migrates from the surface layers to the center of the grain determines the extent of the beneficial effects of parboiling in minimizing milling losses [22]. Several factors, including the levels of minerals in the grain, their solubility during soaking, the ratios of migration, variations in the hydrothermal process, and the parboiled grain’s milling resistance also determine the parboiling effect on grain [13]. 

### 3.5. Percent Contribution to the RDA of B Vitamins and Minerals from the Daily Intake of Different Varieties of Parboiled Polished Rice

The people of Bangladesh usually prefer to consume parboiled polished rice, and the price of rice rises when it is parboiled and polished. According to the Bangladesh Nutrition Survey 2017–2018, an adult consumed 390 g of rice per day [23]. Table 2 shows the percentage contribution to RDA of B vitamins and minerals from the daily intake of different varieties of parboiled polished rice. Daily intake of different varieties of rice covered 64–121% of the RDA of niacin, 24–46% of the RDA of thiamin, 17–40% of the RDA of pyridoxine, 8–25% of the RDA of folate, and 2–7% of the RDA of riboflavin. In the case of minerals, daily intake of different varieties of parboiled polished rice supplied 27–45% of the RDA of Mg, 25–62% of the RDA of Zn, 13–16% of the RDA of Ca, and only 10–14% of the RDA of Fe. BRRI dhan84 contributed a higher percentage of the RDA of all B vitamins and minerals. In contrast to a preceding Indian study, the probable daily dietary intake of calcium and magnesium derived from equal amounts of a local parboiled milled rice variant exhibited notable differences compared with the Bangladeshi BRRI dhan84 variety (calcium: Indian variety 9.8% vs. Bangladeshi BRRI dhan84 13.56%; magnesium: Indian variety 27% vs. Bangladeshi BRRI dhan84 44.54%). However, the contribution of Fe and Zn from the Indian variety was observed to be higher than that from the local variety [22]. As previously elucidated, variations in processing methods and environmental factors contribute to the differences in mineral content among rice samples, consequently impacting their potential contribution to nutrient content. HYVs like BRRI dhan84, which are rich in vitamins and minerals, can play a vital role in combating particular micronutrient deficiencies in the country and attaining better nutrition for the population [6]. The contributions of other processing conditions of different rice varieties to the RDA of B vitamins and minerals are given in Appendix A.

## 4. Conclusions

The present study, with a comprehensive varietal-level nutrient profiling of rice and changes in these nutrient levels due to processing, reveals that the levels of most nutrients were reduced by polishing. The unpolished rice of all varieties contained different nutrients in higher amounts, both in parboiled and unparboiled conditions, compared to polished rice. When parboiling and polishing were both used, macronutrient and vitamin retention were higher in polished parboiled rice than in polished unparboiled rice. The percent contribution to the RDA of vitamins and minerals showed that rice is a key source of vitamins and minerals as a staple food. The findings of the present study will help to assess the fraction of dietary requirements fulfilled according to age, sex, physical activity level, and physiological state. Thus, to combat nutritional problems in Bangladesh, rice could be considered a significant source of various nutrients. Moreover, the findings suggest that the levels of different nutrients in rice varieties are reduced by polishing the rice, while parboiling has a positive effect on nutrient composition. Therefore, to reduce nutrient loss from rice, the relevant industries and consumers should be encouraged to avoid rice that has undergone excessive polishing and recommend consuming unpolished rice in either parboiled or unparboiled conditions. The findings of the present study can help the government in the formulation of policy on the degree of milling required to prevent the loss of nutrients, especially micronutrients, from rice. 

## Figures and Tables

**Figure 1 foods-12-03997-f001:**
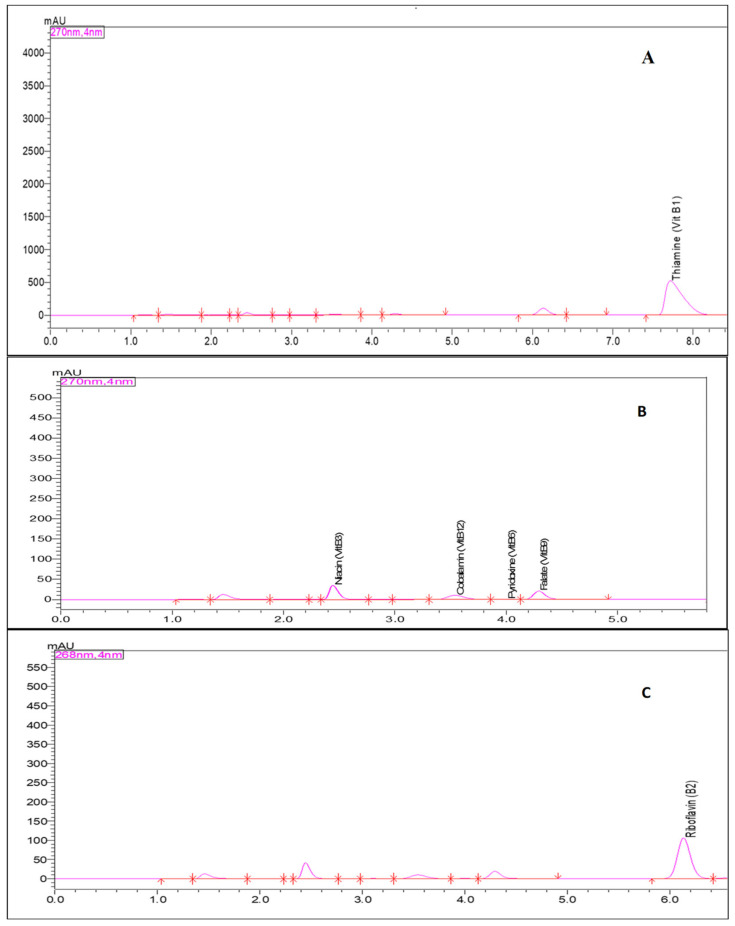
Water-soluble vitamin standards (**A**) thiamin, (**B**) niacin, pyridoxine, and folate, and (**C**) riboflavin are visualized at the common wavelength of 268–270 nm.

**Figure 2 foods-12-03997-f002:**
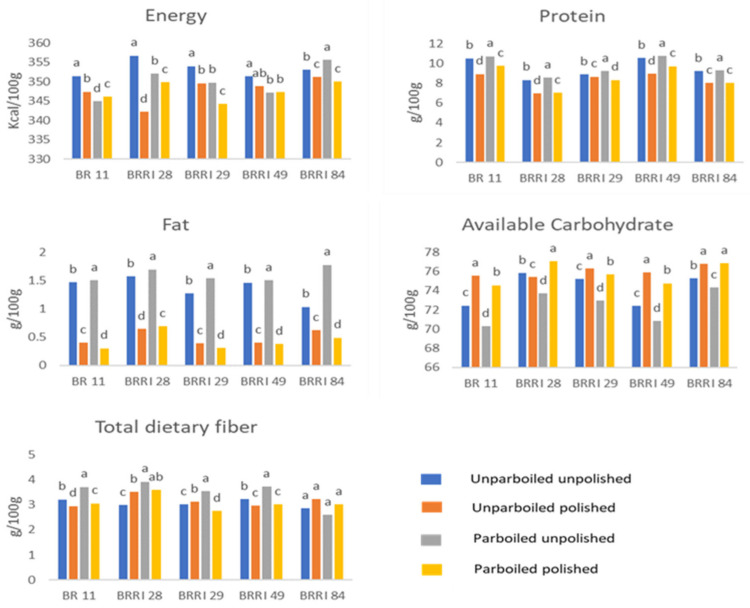
Effects of parboiling and polishing on proximate composition of different rice varieties in Bangladesh. Values with the same superscript letter within the same rice variety do not differ significantly at *p* < 0.05 by one-way ANOVA.

**Figure 3 foods-12-03997-f003:**
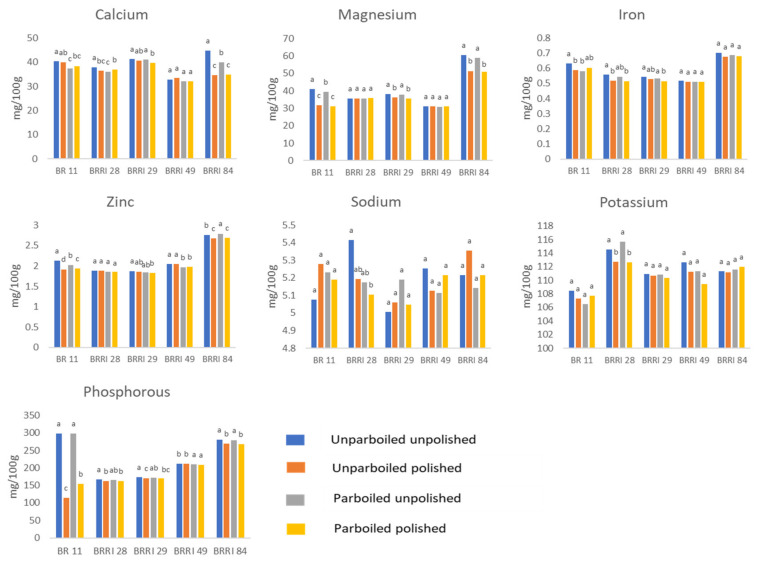
Parboiling and polishing effect on the mineral composition of different rice varieties in Bangladesh. Values with the same superscript letter within the same rice variety do not differ significantly at *p* < 0.05 by one-way ANOVA.

**Figure 4 foods-12-03997-f004:**
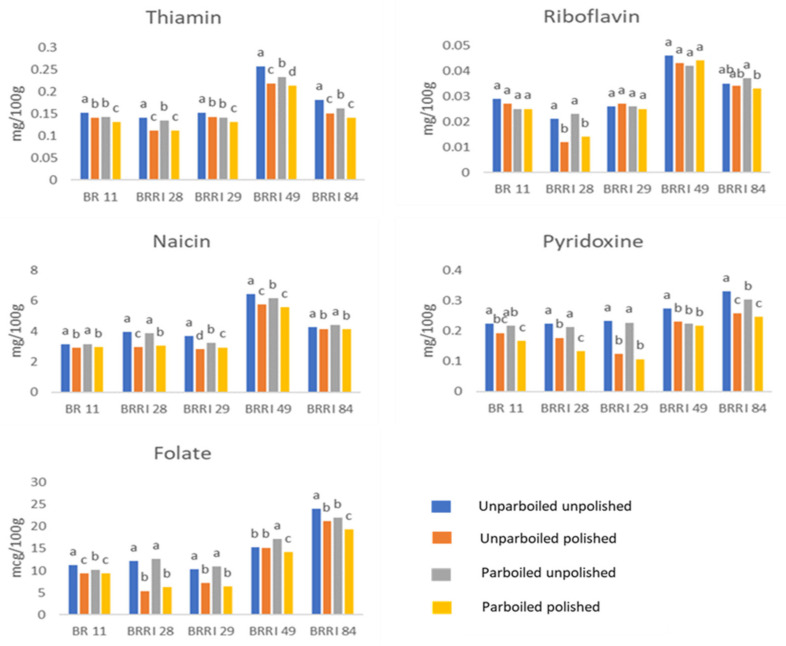
Effect of polishing and parboiling on the B-vitamins composition of different rice varieties in Bangladesh. Values with the same superscript letter within the same rice variety do not differ significantly at *p* < 0.05 by one-way ANOVA.

**Table 1 foods-12-03997-t001:** Calibration standard concentration (mg/Kg).

Elements	Standard 1	Standard 2	Standard 3	Standard 4	r	Wavelength (nm)
Ca	20	100	200	500	0.999	315.8
Mg	4	20	40	100	0.999	285.2
Fe	0.4	2	4	10	0.999	238.2
Zn	0.4	2	4	10	0.999	213.8
Na	20	100	200	500	0.999	589.5
K	20	100	200	500	0.999	766.4
P	20	100	200	500	0.999	213.6

**Table 2 foods-12-03997-t002:** Percent contribution to RDA of B vitamins and minerals by the daily intake of different varieties of parboiled polished rice.

Nutrients (% Contribution of RDA)	BR11	BRRI dhan28	BRRI dhan29	BRRI dhan49	BRRI dhan84
Thiamin	28.5	24.1	28.5	30.6	46.1
Riboflavin	4.0	2.2	3.9	5.1	6.9
Niacin	63.8	66.4	63.6	89.2	120.8
Pyridoxine	27.1	21.7	17.3	35.2	40.1
Folate	12.2	8.2	8.4	18.4	25.1
Ca	15.0	14.4	15.5	12.5	13.6
Mg	27.5	31.8	31.5	27.4	44.5
Fe	12.4	10.6	10.6	10.5	13.9
Zn	44.5	24.5	42.0	45.3	61.8

## Data Availability

The datasets generated for this study are available on request to the corresponding author.

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
