# Peer review of "Variations in the Major Nutrient Composition of Dominant High-Yield Varieties (HYVs) in Parboiled and Polished Rice of Bangladesh"

_foods, 2023, doi:10.3390/foods12213997_

Round 1
Reviewer 1 Report
Comments and Suggestions for Authors
Comments to authors
The study was well-designed, and the results are detailed enough to examine the mechanisms behind the reported trends. There is a need to justify the following comments, in order to understand their practicability for the stated objectives.
Please add the background of the study within 2-3 sentences at the start of the abstract.
The abstract is too long. Please concise the abstract and only provide the main findings.
The introduction should include research justification of major nutrient composition affected by polished and boiled rice from previous studies.
The aims of the study are not clear from Lines no 77-81. Please further elaborate on the aims and give more details
Please provide specific model of “Satake Rice mill” in section 2.1
Write down the abbreviation of “minutes” to “min” in line no 94 and throughout the manuscript.
Enlarge the font size in Figure 1 to make values more visible.
Please mention what kind of ANOVA performed for analysis in section 2.5? Is it one way or two way?
The results and discussion do not go further than expected. It lacks a deepening. The results need proper discussion and comparison with previous similar studies to highlight the usefulness of the study.
Compare the results of section 3.5 with previous studies.
The novelty of the manuscript is questionable as already a lot of literature has been published. Please highlight the novelty of the manuscript
The conclusion needs to be improved. The conclusion is too general. What were the outcomes and prospects are not clear? The authors should justify elaborately how the current study is different from previous studies and useful to the scientific community.
Comments on the Quality of English LanguageMinor improvements in the English language are required.
Author Response
Author's response to the Reviewer 1:
The study was well-designed, and the results are detailed enough to examine the mechanisms behind the reported trends. There is a need to justify the following comments, in order to understand their practicability for the stated objectives.
Reviewer’s comment: Please add the background of the study within 2 sentences at the start of the abstract.
Author’s response: We have added two sentences in the abstract section highlighting the background of the study. See the sentences in lines 1-16.
Reviewer’s comment: The abstract is too long. Please concise the abstract and only provide the main findings.
Author’s response: We have rewritten the abstract following the suggestion of the two reviewers.
Reviewer’s comment: The introduction should include research justification of major nutrient composition affected by polished and boiled rice from previous studies.
Author’s response: Thank you for your valuable suggestion. Research justification has been incorporated as follows:
“To ascertain the nutritional composition of food, commonly referenced databases such as the USDA National Nutrient Database and national equivalents like the Food Composition Table for Bangladesh (FCTB), are frequently employed [13]. Nevertheless, these databases often lack consideration for varietal disparities. It is anticipated that various HYVs may exhibit distinct responses to analogous processing treatments, such as polishing and parboiling. Consequently, for a more comprehensive understanding, it is imperative to account for varietal differences in order to achieve precision in nutrient intake assessments.” See the lines 82-90.
Reviewer’s comment: The aims of the study are not clear from Lines no 77-81. Please further elaborate on the aims and give more details.
Author’s response: Here in this paragraph, we mentioned the objective of the study and then highlighted the expected policy implication of the study findings. However, following your suggestion, we have revised the sentences. Please see the lines 91-101.
Reviewer’s comment: Please provide a specific model of “Satake Rice mill” in section 2.1
Author’s response: We have provided the specific model of “Satake Rice mill” in section 2.1. See line 119 in the revised manuscript.
Reviewer’s comment: Write down the abbreviation of “minutes” to “min” in line no 94 and throughout the manuscript.
Author’s response: Thank you for your suggestion. We have changed the abbreviation of “minutes” to “min” throughout the manuscript.
Reviewer’s comment: Enlarge the font size in Figure 1 to make values more visible.
Author’s response: We have edited the Figure. Please see Figure 1, line 162.
Reviewer’s comment: Please mention what kind of ANOVA was performed for analysis in section 2.5. Is it one-way or two-way?
Author’s response: Thank you for your incisive comment. We have mentioned that we have performed a one-way ANOVA analysis in section 2.5. Please see the lines 196-198 in the revised manuscript.
Reviewer’s comment: The results and discussion do not go further than expected. It lacks a deepening. The results need proper discussion and comparison with previous similar studies to highlight the usefulness of the study.
Author’s response: Comparison with earlier studies and some extended discussion points have been added throughout the manuscript. Please see the lines 223-241, 266-269, 283-285, 313-321, 379-393
Reviewer’s comment: Compare the results of section 3.5 with previous studies.
Author’s response: The following excerpt has been incorporated:
“In contrast to a preceding Indian study, the probable daily dietary intake of calcium and magnesium derived from the equal amount of local parboiled milled rice variant exhibited notable distinctions when juxtaposed with the Bangladeshi BRRI 84 variety (Calcium: Indian variety 9.8% vs. Bangladeshi BRRI 84 13.56%; Magnesium: Indian variety 27% vs. Bangladeshi BRRI84 44.54%). However, the contribution of Fe and Zn from the Indian variety was observed to be higher than the local variety.[23]. As previously elucidated, variations in processing methods and environmental factors contribute to the divergence in mineral content among rice samples, consequently impacting their potential contribution to nutrient content. Incidentally, HVYs like BRRI 84 that are rich in vitamins and minerals can play a vital role in combating particular micronutrient deficiencies in the country and attaining better nutrition of the population [6].” See the lines 379-393.
Reviewer’s comment: The novelty of the manuscript is questionable as already a lot of literature has been published. Please highlight the novelty of the manuscript.
Author’s response: Thank you for the feedback. The manuscript introduces novelty through a comprehensive approach, distinct from earlier studies. Unlike previous works, this study explored new rice varieties, providing extensive (18) nutrient profiling encompassing proximate, vitamins (thiamine, riboflavin, niacin, pyridoxine, folate), and minerals (calcium, magnesium, iron, zinc, sodium, potassium, phosphorus) content which have a public health problem among the population of Bangladesh. Significantly, previous studies lacked the analysis presented here, as they did not calculate the daily nutrient contribution (%RDA) specifically from rice by using the per capita consumption. Furthermore, our study uniquely examines the combined effects of variety, milling, and parboiling, offering a more inclusive understanding of these factors collectively and generating evidence for the policy formulation for millers on polishing to prevent the loss of nutrients, especially micronutrients. This multifaceted approach contributes to the novelty of the manuscript, filling critical gaps in the existing literature. Please see the lines 81-101.
Reviewer’s comment: The conclusion needs to be improved. The conclusion is too general. What were the outcomes and prospects that were not clear? The authors should justify elaborately how the current study is different from previous studies and useful to the scientific community.
Author’s response: We have revised the conclusion section and described the outcomes and future prospects of the study. See the lines 398-416 in the conclusion section.
Reviewer 2 Report
Comments and Suggestions for Authors
A well-written paper
a printing error in figure 2, 3 and 4 should be corrected. The text after the yellow bar should be Parboiled polished
- Minor correction of English is required. The paper can then be accepted. However:
- The impact of the paper will increase when – in addition to the presented results - a comparison is made with similar results of ‘classical’ rice varieties in Bangladesh.
Minor correction of English is required.
Author Response
Author's response to Reviewer 2:
Comments and Suggestions for Authors
A well-written paper
Reviewer’s comment: A printing error in figures 2, 3, and 4 should be corrected. The text after the yellow bar should be Parboiled polished.
Author’s response: Thank you for pointing out the error in Figures 2,3 and 4. We have revised the figures and addressed the error. See figures 2-4 in the revised manuscript, lines 251, 299, 349.
Reviewer’s comment: Minor correction of English is required.
Author’s response: We have checked the grammatical errors, punctuation, and spelling in our revised manuscript.
Reviewer’s comment: The paper can then be accepted. However: The impact of the paper will increase when – in addition to the presented results - a comparison is made with similar results of ‘classical’ rice varieties in Bangladesh.
Author’s response: Thank you for your suggestion. We have compared the findings with previous studies conducted in Bangladesh observing indigenous varieties and other countries. Please see the comparison in lines 223-241, 266-269, 283-285, 313-321, 379-393.
Reviewer 3 Report
Comments and Suggestions for Authors
The study is full of information, exploring the variability in nutrient composition of five dominant rice varieties in Bangladesh. However, it requires an accurate review, mainly regarding the section of Results and Discussions more concentrated on the description than on discussion. Therefore, the authors are strongly encouraged to improve the way of presenting the results and to implement the discussion by adding other references. Also, the conclusions have to emphasize the practical applications of this research. Review English and grammar (for example, English would not capitalise after a colon).
Below some recommendations.
Title should be revised as “Variation of major nutrient composition of dominant high-yield varieties (HYVs) in parboiled and polished rice of Bangladesh”.
Abstract is to be revisited following the notes below. Furthermore, the authors should evidence the mean results, suggesting implications or applications of the research you discuss in the paper.
Lines 16-21. The authors refer to the recommended methods for vitamins and minerals without data. After, considering other proximate parameters as energy, protein etc, the range were reported with no methodologies. Please, revise the abstract carefully.
Lines 21-23. ‘While polishing and parboiling were undertaken together, macronutrient and vitamin retention were higher 22 than polished non-parboiled rice’. Check the meaning of the sentence.
Introduction
Lines 59-60. “Parboiling is another important step which includes:” change with “Parboiling is another important processing technique, which includes three additional steps to rice industrialization:”.
Lines 72-73. The name of varieties (BR11, BRRI dhan28, BRRI dhan29, BRRI dhan49, 72 and BRRI dhan84) has just been reported in materials and methods, so in this section can be deleted.
Lines 77-81. Carefully check this period, distinguish between the aim of the work and the conclusions and future perspectives.
Materials and Methods
Lines 90-101. The text should clearly highlight the procedures to obtain the four types of samples for each variety. For line 91, delete the text “in the first sentence of rice parboiling” and add before “300g” the sentence “To obtain parboiled grains” changing the verb ‘was’ with ‘were’.
Lines 109-110. Specify which compound were used to estimate the available carbohydrate.
Lines 114-130. Check the text. Generally, a reference method was reported at the beginning of the determination. Furthermore, if the method used for the determination of B-group vitamins has been modified, detail it, otherwise it is apposite to summarize the method.
Lines 140-141. It is appropriate to report the unit of measurement for minerals only once.
Line 159. Please explain why calibration standards of minerals were expressed as mg/kg while the samples as mg/100g (see lines 140-141 and related Tables)?
Lines 162-163. As reported in the text, data have been expressed as mean ± standard deviation. Therefore, use this approach also for Figures 2-3.
Results and discussion
Lines 169-172. It is more appropriate for this period to be reported in section 2.1 of Materials and Methods.
Line 179. No results or comments have been followed Table 2.
Line 180. Check the range of protein (8-11.92 g/100g).
Line 181. Please, insert the name of Figure. Furthermore, no mention in the text to the “Available Carbohydrate” has been reported.
Line 182. “The combined effect of polishing and parboiling: Regardless of the parboil”. Please, correct your punctuation errors.
Lines 182-187. Rewrite this text in a scientific language. In line 183, please delete 10% as ‘within’ means ‘not be more than’.
Line 188. Specify in material and methods that brown rice means ‘non-parboiled unpolished’.
Lines 188-199. It’s better to comment the parameters in the sequence in which they have been tabulated. Check the range of proximate composition (Suppl Table 4). What do the authors mean by such an expression ‘a certain level of decrease in protein and fat’?
Lines 200-208. Please, clarify this period in particular the protein levels (lines 202 and 204).
Lines 213-221. The expressions ‘where the effect of polishing was low’ or ‘overall nutrient retention’ are too generic. For which parameters?
Lines 225-226. ‘In contrast, parboiling increased the ash content in rice’ with values, most of the time, higher than the brown one. Check. Are there any legal limits?
Lines 227-228. Is this the case for all the analyzed parameters? Move reference to Supplementary Tables 7 and 8 after ‘polishing’. Please, verify the correct numbering of the Supplementary Tables whose numerical sequence must be progressive. Instead, the authors move from Suppl Table 4 to Suppl Tables 7-8. Moreover, in the percentage of nutrient loss respect to (??? specify) also the combined effect of polishing and parboiling should be considered.
Line 232. Figure 3 regards vitamin composition of rice varieties, nor mineral contents. Check.
Lines 235-254. Re-visit this paragraph as the authors should outline the most significant differences, among the different processing, for the vitamin content. The authors stated that ‘Therefore, nutrient loss during parboiling is lesser than compared to polishing’, what is the benefit of this combined technique?
Paragraph 3.4. It would be appropriate to move this paragraph to the previous paragraph, after ash (i.e., a measure of total mineral content in cereals) discussion reported at the end of paragraph 3.2..
Lines 256-257 Figure 4 presents the effect of different processing conditions on the mineral contents of….
Lines 266-268. Check the text ‘The level of Ca, Mg, K, 266 and Fe sharply decreased after polishing compared to unpolished non-parboiled 267 rice (Supplementary Table 7)” as it does not correspond to Suppl, Table 7 “non-parboiled vs. parboiled”.
Lines 276-277. Check Tukey in Figure 4 and Suppl Table 6, in particular for Na.
Lines 293-294. Re-formulate the sentence.
Lines 293-305. All this period, interpreted as a conclusion of this paragraph, in part is referred to general consideration for all nutrients and in part to minerals. It should be re-elaborate emphasizing for minerals the processing technique with minor losses and demanding to the conclusion general considerations on all nutrients.
Line 318. Substitute ‘other forms of different rice varieties’ with ‘the other processing conditions of different rice varieties’.
Line 328. Please, insert a sentence on the combination of polishing and parboiling techniques.
Lines 333-334. Conclusion (i.e., to encourage people to consume brown rice) is the most simplistic and does not take into account the food chain, from field to processing. Furthermore, how is it possible to take advantage from the information reported in this paper? As nutrient loss during parboiling is lesser than compared to polishing, which are the future perspectives for industries and, consequently for consumers?
Reference n. 15.
Delete point between 1 and 5.
Figures 2-3. Check carefully the legend as the ‘non-parboiled polished’ has been reported twice.
Figure 4. Uniform Figure 4 to the others, also by placing the correct orientation of the Tukey letters.
Table 2. Being data reported in Table 2 as percent contribution to RDA, it is enough express data with only one decimal place.
Comments on the Quality of English LanguageReview English and grammar
Author Response
Author's response to Reviewer 3:
Comments and Suggestions for Authors
The study is full of information, exploring the variability in nutrient composition of five dominant rice varieties in Bangladesh. However, it requires an accurate review, mainly regarding the section on Results and Discussions more concentrated on the description than on the discussion. Therefore, the authors are strongly encouraged to improve the way of presenting the results and to implement the discussion by adding other references. Also, the conclusions have to emphasize the practical applications of this research. Review English and grammar (for example, English would not capitalize after a colon). Below are some recommendations.
Reviewer’s comment: The title should be revised as “Variation of the major nutrient composition of dominant high-yield varieties (HYVs) in parboiled and polished rice of Bangladesh”.
Author’s response: Thank you for your suggestion. Following your suggestion, we have revised the title to “Variation of the major nutrient composition of dominant high-yield varieties (HYVs) in parboiled and polished rice of Bangladesh”. See the lines 2-4.
Reviewer’s comment: The abstract is to be revisited following the notes below. Furthermore, the authors should evidence the mean results, suggesting implications or applications of the research you discuss in the paper.
Author’s response: We have revised the abstract section following your suggestion. We have also added a few points on the implications or applications of the findings. See the line 13-37 in the abstract.
Reviewer’s comment: Lines 16-21. The authors refer to the recommended methods for vitamins and minerals without data. After, considering other proximate parameters such as energy, protein, etc., the range was reported with no methodologies. Please, revise the abstract carefully.
Author’s response: Thank you for your suggestions. We have briefly mentioned the methods for estimating the proximate and micronutrients and revised the manuscript. See the line 21-26 in the abstract.
Reviewer’s comment: Lines 21-23. ‘While polishing and parboiling were undertaken together, macronutrient and vitamin retention were higher than polished non-parboiled rice’. Check the meaning of the sentence.
Author’s response: We have revised the sentence as “While polishing and parboiling were undertaken together, macronutrients and vitamins retention was higher in parboiled unpolished rice as compared to polished un-parboiled rice” to make it more understandable to the readers. Please see the lines 30-31 in the abstract.
Reviewer’s comment: Introduction Lines 59-60. “Parboiling is another important step which includes:” change with “Parboiling is another important processing technique, which includes three additional steps to rice industrialization:”.
Author’s response: Thank you for your suggestion. We have revised the sentence as “Furthermore, parboiling is another important processing technique, which includes three additional steps to rice industrialization: soaking, pressure steaming, and drying before dehusking”. See the line 70-72 in the introduction section.
Reviewer’s comment: Lines 72-73. The name of varieties (BR11, BRRI dhan28, BRRI dhan29, BRRI dhan49, 72, and BRRI dhan84) has just been reported in materials and methods, so this section can be deleted.
Author’s response: We have deleted the name of the rice varieties to avoid redundancy and keep the sentence as “The present study aimed to investigate the impact of polishing and parboiling on the nutrient composition of the five HYVs of rice in Bangladesh”. See the line 93-95 in the introduction section.
Reviewer’s comment: Lines 77-81. Carefully check this period, and distinguish between the aim of the work and the conclusions and future perspectives.
Author’s response: Here in this paragraph, we mentioned the objective of the study and then highlighted the expected policy implication of the study findings. However, following your suggestion, we have revised the sentences as “It is anticipated that the study's findings, which highlight potential variations in important macronutrients and micronutrients among the rice varieties processed using various ways, would help with the establishment of the appropriate policies to prevent the loss of nutrients.” See the lines 103-126.
Reviewer’s comment: Materials and Methods Lines 90-101. The text should clearly highlight the procedures to obtain the four types of samples for each variety. For line 91, delete the text “in the first sentence of rice parboiling” and add before “300g” the sentence “To obtain parboiled grains” changing the verb ‘was’ with ‘were’.
Author’s response: We appreciate your valuable suggestion. We have revised the section on “Sample collection and processing” according to your suggestion. We have added the procedures to obtain the four types of samples (non-parboiled unpolished, non-parboiled polished, parboiled unpolished, and par-boiled polished) for each variety. See the lines 103-126 in the “Sample collection and processing” section.
Reviewer’s comment: Lines 109-110. Specify which compounds were used to estimate the available carbohydrate.
Author’s response: Thank you for your suggestion. We have mentioned the compounds that we used to estimate the available carbohydrates. We have added it as “The content of available carbohydrate in the rice samples was calculated by difference: by subtracting the sum percentage of moisture, protein, fat, ash, total dietary fiber”. See lines 137-139 in the revised manuscript.
Reviewer’s comment: Lines 114-130. Check the text. Generally, a reference method was reported at the beginning of the determination. Furthermore, if the method used for the determination of B-group vitamins has been modified, detail it, otherwise, it is apposite to summarize the method.
Author’s response: Thank you for your suggestion. We have mentioned it in our revised manuscript. See the lines 159-161.
Reviewer’s comment: Lines 140-141. It is appropriate to report the unit of measurement for minerals only once.
Author’s response: We have revised the sentence as “Calcium (Ca) and Magnesium (Mg), Iron (Fe), Zinc (Zn), Sodium (Na), Potassium (K), and Phosphorus (P) were measured at mg/Kg (ppm) units by Shimadzu ICP-OES 9820 and later were converted at mg/100g as an edible portion.”. See the lines 171-173 in the manuscript.
Reviewer’s comment: Line 159. Please explain why calibration standards of minerals were expressed as mg/kg while the samples as mg/100g (see lines 140-141 and related Tables).
Author’s response: Calcium (Ca) and Magnesium (Mg), Iron (Fe), Zinc (Zn), Sodium (Na), Potassium (K), and Phosphorus (P) were measured at mg/Kg (ppm) units by Shimadzu ICP-OES 9820 and later were converted at mg/100g as serving size. See the lines 171-173.
Reviewer’s comment: Lines 162-163. As reported in the text, data have been expressed as mean ± standard deviation. Therefore, use this approach also for Figures 2-3.
Author’s response: We appreciate your suggestion. We presented the data as mean ± standard deviation in supplementary Table 4, 7, 8 with ANOVA test results. We couldn’t present the mean, or standard deviation value in Figures 2-4 to avoid clumsiness and make the Figure more understandable to the readers. However, the readers will find the mean, SD values as they have been added as supplementary Tables 4, 7, 8.
Results and discussion
Reviewer’s comment: Lines 169-172. It is more appropriate for this period to be reported in section 2.1 of Materials and Methods.
Author’s response: We have revised the sentences in the result section 3.1 and transferred the information in section 2.1 of Materials and Methods. See lines 106-110 in the method section 2.1 and lines 202-204 in the result section 3.1.
Reviewer’s comment: Line 179. No results or comments have been followed in Table 2.
Author’s response: Thank you for your observation. We have revised the sentence. It was actually Figure 2 which had been mistakenly reported as Table 2. See the correction in line 209.
Reviewer’s comment: Line 180. Check the range of protein (8-11.92 g/100g).
Author’s response: We have checked the range of protein contents and revised the values. See the correction in the line 210.
Reviewer’s comment: Line 181. Please, insert the name of the Figure. Furthermore, no mention in the text to the “Available Carbohydrate” has been reported.
Author’s response: We have revised those sentences and mentioned available carbohydrates throughout the manuscript. Moreover, we have also inserted the name of the figure and table. See the line 219 in the result section.
Reviewer’s comment: Line 182. “The combined effect of polishing and parboiling: Regardless of the parboil”. Please, correct your punctuation errors.
Author’s response: We have revised the sentence to avoid punctuation errors.
Reviewer’s comment: Lines 182-187. Rewrite this text in a scientific language. In line 183, please delete 10% as ‘within’ means ‘not be more than’.
Author’s response: Thank you for your suggestion. We have revised the sentences as per suggestion. See the revision in the lines 212-213.
Reviewer’s comment: Line 188. Specify in material and methods that brown rice means ‘non-parboiled unpolished’.
Author’s response: In the material and method section, we have mentioned that brown rice means unpolished rice. See lines 218-219 in the material and method section.
Reviewer’s comment: Lines 188-199. It’s better to comment on the parameters in the sequence in which they have been tabulated. Check the range of proximate composition (Suppl Table 4). What do the authors mean by such an expression ‘a certain level of decrease in protein and fat’?
Author’s response: Thank you for your suggestion. We have checked the range of proximate composition (Suppl Table 4). We presented the nutrients in the sequence in which they have been tabulated. We have also revised the sentences. See the lines 219-241.
Reviewer’s comment: Lines 200-208. Please, clarify this period in particular the protein levels (lines 202 and 204).
Author’s response: Thank you for your suggestion. The issue has been addressed as follows:
“Previous research yielded comparable results regarding milling effects on Brazilian rice samples. Specifically, brown rice exhibited a protein content of 6.85%, which decreased to 6.66% post-milling. Another study showed the protein content in brown rice (9.2%) decreased with increasing DOM indicating a decline in protein content from the rice surface to the endosperm; furthermore, the continued decrease after bran removal implies non-uniform distribution, with lower content in the core endosperm compared to the outer layers [10].” See the lines 219-229.
Reviewer’s comment: Lines 213-221. The expressions ‘where the effect of polishing was low’ or ‘overall nutrient retention’ are too generic. For which parameters?
Author’s response: We have revised the sentence as “However, among the rice varieties, parboiling caused high nutrient retention compared to un-parboiled polished rice (Figure 2).”
Reviewer’s comment: Lines 227-228. Is this the case for all the analyzed parameters? Move reference to Supplementary Tables 7 and 8 after ‘polishing’. Please, verify the correct numbering of the Supplementary Tables whose numerical sequence must be progressive. Instead, the authors move from Suppl Table 4 to Suppl Tables 7-8. Moreover, in the percentage of nutrient loss with respect to (??? specify) also the combined effect of polishing and parboiling should be considered.
Author’s response: We have corrected the numbering of the supplementary tables.
Reviewer’s comment: Line 232. Figure 3 regards the vitamin composition of rice varieties, nor mineral contents. Check.
Author’s response: Thank you for your observation. We have checked the sentence and revised it accordingly. See the line 325-326.
Reviewer’s comment: Lines 235-254. Re-visit this paragraph as the authors should outline the most significant differences, among the different processing, for the vitamin content. The authors stated that ‘Therefore, nutrient loss during parboiling is lesser than compared to polishing’, what is the benefit of this combined technique?
Author’s response: Thank you for the feedback, the issue has been addressed as follows:
“However, when the processes of polishing and parboiling are concurrently applied, the collective mineral retention surpasses that of exclusively polished rice but falls short of the levels observed in solely parboiled rice. Consequently, polishing, a practice undertaken to enhance consumer appeal, while combined with parboiling, augments nutrient availability.” See the lines 272-277.
Reviewer’s comment: Paragraph 3.4. It would be appropriate to move this paragraph to the previous paragraph, after the ash (i.e., a measure of total mineral content in cereals) discussion reported at the end of paragraph 3.2.
Author’s response: Thank you for your comment. We have rearranged the paragraph in our revised manuscript.
Reviewer’s comment: Lines 256-257 Figure 4 presents the effect of different processing conditions on the mineral contents of…
Author’s response: We have revised the sentence as “Figure 4 presents the effect of different processing conditions on the mineral contents of different dominant HY rice varieties in Bangladesh.” See the corrections in the lines 279-280.
Reviewer’s comment: Lines 266-268. Check the text ‘The level of Ca, Mg, K, and Fe sharply decreased after polishing compared to unpolished non-parboiled rice (Supplementary Table 7)” as it does not correspond to Suppl, Table 7 “non-parboiled vs. parboiled”.
Author’s response: Thank you for spotting. See the corrections in lines 287-289.
Reviewer’s comment: Lines 276-277. Check Tukey in Figure 4 and Suppl Table 6, in particular for Na.
Author’s response: We checked the analysis and Tucky letters reported in Figure 3 and Supplementary Table 8. The values were reported correctly.
Reviewer’s comment: Lines 293-294. Re-formulate the sentence.
Author’s response: We have revised the sentence as “In this study, parboiling (without polishing) had substantially higher nutrient retention than polishing.” See the corrections in the lines 354-355.
Reviewer’s comment: Line 318. Substitute ‘other forms of different rice varieties’ with ‘the other processing conditions of different rice varieties’.
Author’s response: We have revised the sentence as “The contribution of the other processing conditions of different rice varieties to the RDA of B vitamins and minerals is given in Supplementary Table 9.” See the correction in lines 391-393.
Reviewer’s comment: Line 328. Please, insert a sentence on the combination of polishing and parboiling techniques.
Author’s response: We have added a sentence regarding the effect of the combination of polishing and parboiling techniques on the nutrient content of the rice samples. See the lines 402-404 in the conclusion section.
Reviewer’s comment: Lines 333-334. Conclusion (i.e., to encourage people to consume brown rice) is the most simplistic and does not take into account the food chain, from field to processing. Furthermore, how is it possible to take advantage of the information reported in this paper? As nutrient loss during parboiling is lesser than compared to polishing, what are the future perspectives for industries and, consequently for consumers?
Author’s response: Thank you for your comments. We have revised the conclusion section highlighting the future perspectives for industries and consumers as “The findings suggest that the content of different nutrients in rice varieties is reduced due to the polishing of the rice and parboiling has little effect on nutrient composition. Therefore, to reduce the nutrient loss of rice, the respective industries and the consumers should be encouraged to avoid polishing and consume unpolished rice in either parboiled or non-parboiled conditions.” See the lines in the conclusion section 409-414.
Reviewer’s comment: Reference n. 15. Delete points between 1 and 5.
Author’s response: We have deleted the point between 1 and 5 in reference number 15. See the correction in line 443 in the references.
Reviewer’s comment: Figures 2-3. Check carefully the legend as the ‘non-parboiled polished’ has been reported twice.
Author’s response: Thank you for your observation. We have made necessary corrections in the legend of Figures 2, 3, and 4.
Reviewer’s comment: Figure 4. Uniform Figure 4 to the others, also by placing the correct orientation of the Tukey letters.
Author’s response: We have changed the orientation of the Tukey letters in Figure 3 to make it uniform with Figures 2 and 4.
Reviewer’s comment: Table 2. Being data reported in Table 2 as a percent contribution to RDA, it is enough to express data with only one decimal place.
Author’s response: Thank you for your suggestion. We have kept the values in one decimal place. See the revision in Table 2 of our revised manuscript.
Round 2
Reviewer 3 Report
Comments and Suggestions for Authors
Although the authors have responded to all revisions, a thorough review of the English is needed in all the text, particularly in the added sentences. Below some other suggestions:
Lines 22-26. Change sentences as “moisture, ash, fat, and total dietary fiber (TDF) were determined gravimetrically, according to the ' AOAC Official Methods, protein by Kjeldahl method, B-group vitamins using Ultra Pressure Liquid Chromatography, and mineral contents by ICP-OES”.
Lines 92-93. Please, delete the name of the rice varieties as just reported in the previous revision.
Lines 159-161. The authors refer that ‘The method is described for the determination of thiamin and riboflavin’, and the other B-group vitamins? Further, move in the right place the revised sentence, not at the end, as previously suggested.
Line 225. Change ‘post-milling’ with ‘after milling’.
Lines 242-244. It is inappropriate the verb ‘caused’ as high nutrient retention is a positive effect.
Comments on the Quality of English LanguageThe manuscript needs detailed revision for language and grammar
Author Response
Although the authors have responded to all revisions, a thorough review of the English is needed in all the text, particularly in the added sentences. Below some other suggestions:
Reviewer’s comments: Lines 22-26. Change sentences as “moisture, ash, fat, and total dietary fiber (TDF) were determined gravimetrically, according to the ' AOAC Official Methods, protein by Kjeldahl method, B-group vitamins using Ultra Pressure Liquid Chromatography, and mineral contents by ICP-OES”.
Authors’ response: Thank you for your suggestion. We have revised the sentence. See the lines 22-24.
Reviewer’s comments: Lines 92-93. Please, delete the name of the rice varieties as just reported in the previous revision.
Authors’ response: We have deleted the name of the varieties of rice as in lines 89-90.
Reviewers’ comment: Lines 159-161. The authors refer that ‘The method is described for the determination of thiamin and riboflavin’, and the other B-group vitamins? Further, move in the right place the revised sentence, not at the end, as previously suggested.
Authors’ response: Thank you for your concern and suggestion. The method was used for the determination of thiamin and riboflavin. We have revised the sentence as you suggested. See the lines 156-157
Reviewer’s comment: Line 225. Change ‘post-milling’ with ‘after milling’.
Authors’ response: We have changed the term post-milling to after milling. See the correction in line 222.
Reviewers’ comment: Lines 242-244. It is inappropriate the verb ‘caused’ as high nutrient retention is a positive effect.
Authors’ response: Thank you for spotting the error. We have changed the verb as “parboiling (without polishing) leads to high nutrient retention”. See the correction in lines 239-241.